# Identification of asymptomatic *Entamoeba histolytica* infection by a serological screening test: A cross-sectional study of an HIV-negative men who have sex with men cohort in Japan

Yasuaki Yanagawa[1,2], Rieko Shimogawara[2], Misao Takano[1], Takahiro Aoki[1], Daisuke Mizushima[1], Hiroyuki Gatanaga[1,3], Yoshimi Kikuchi[1], Shinichi Oka[1,3], Kenji Yagita[2], Koji Watanabe[1,2]*

1 AIDS Clinical Center, National Center for Global Health and Medicine, Tokyo, Japan, 2 Department of Parasitology, National Institutes of Infectious Diseases, Tokyo, Japan, 3 Joint Research Center for Human Retrovirus Infection, Kumamoto University, Kumamoto, Japan

* kwatanab@acc.ncgm.go.jp

## Abstract

### Background

Amebiasis, caused by *Entamoeba histolytica*, is spreading in developing countries and in many developed countries as a sexually transmitted infection. Here, we evaluated the efficacy of serological screening to identify asymptomatic *E. histolytica* infection as a potential epidemiological control measure to limit its spread.

### Methodology/Principal findings

This cross-sectional study was carried out between January and March 2021 in an HIV-negative men who have sex with men (MSM) cohort at the National Center for Global Health and Medicine. Serological screening was performed using a commercially available ELISA kit. For seropositive individuals, we performed stool polymerase chain reaction (PCR) to determine current *E. histolytica* infection. We performed *E. histolytica* serological screening of 312 participants. None had a history of *E. histolytica* infection prior to the study. The overall *E. histolytica* seropositivity was 6.7% (21/312), which was similar to that found by the rapid plasma reagin test (17/312). We identified current infection in 8 of 20 seropositive participants (40.0%) by stool PCR.

### Conclusions/Significance

Our serological screening approach constitutes a potentially practical epidemiological strategy. Active epidemiological surveys, in combination with an effective screening strategy for asymptomatically infected individuals, should be applied to help reduce sexually transmitted *E. histolytica* infections.

**Data Availability Statement:** All relevant data are within the manuscript and its Supporting Information files.

**Funding:** This work was supported by the Japan Agency for Medical Research and Development (AMED) under grant number JP20fk0108138 and by a grant from the National Center for Global Health and Medicine under grant number 21A1002 to KW. The funders had no role in study design, data collection and analysis, decision to publish, or preparation of the manuscript.

**Competing interests:** The authors have declared that no competing interests exist.

## Author summary

Amebiasis, caused by *Entamoeba histolytica*, is now spreading not only in developing countries, but also in many of developed countries. Unlike the situation in developing countries, transmission occurs directly from one infected person to another via sexual contact, called sexually transmitted *E. histolytica* infection. Furthermore, most cases of *E. histolytica* infection are asymptomatic, who can be a reservoir for sexual transmission in the community. Cost-effective epidemiological strategy is urgently needed. Hereby, we performed a serological test for 312 "asymptomatic" HIV-negative men who have sex with men to assess the effective screening method for *E. histolytica* infection. We identified 21 seropositive samples (6.7% of seropositivity, 21/312), in which relatively high seropositivity to *E. histolytica* was seen among the participants with positive serology for *Treponema pallidum* hemagglutination (TPHA) or hepatitis B core antibody (HBcAb). Finally, we identified current infection (asymptomatic *E. histolytica* infection) in 8 out of 20 stool sampling cases (40.0%) by polymerase chain reaction. Our serological screening assay provides a potentially practical epidemiological strategy. Active epidemiological survey, in combination with the effective screening strategy for asymptomatically infected individuals are considered for the future control of sexually transmitted *E. histolytica* infection.

## Introduction

Amebiasis is an intestinal protozoa infection caused by *Entamoeba histolytica*, which is the second most common cause of parasite-related deaths worldwide and is particularly found in developing countries [1]. It is also a growing concern in some developed countries in East Asia and Europe, where *E. histolytica* infection is increasingly prevalent as a sexually transmitted infection [2–4]. In Japan, men who have sex with men (MSM) are reported to be at especially high risk for sexually transmitted *E. histolytica* infection [5,6]. Life-threatening cases of *E. histolytica* infection are accumulating in these countries [7–9]. Moreover, many of these cases were not diagnosed until autopsy [9]. This is likely because *E. histolytica* infection is a neglected disease; thus, it is rarely suspected in the clinical setting when a patient has acute abdominal symptoms. Hence, an effective epidemiological strategy to reduce *E. histolytica* infections is urgently needed. In developing countries, transmission typically occurs as a result of unsanitary conditions; however, transmission can also directly occur between people through sexual contact [10]. Furthermore, most cases of *E. histolytica* are asymptomatic [11]. Indeed, seroprevalence data has shown that asymptomatic infection occurs among sexually active individuals [12,13] who act as a reservoir for sexual transmission. Polymerase chain reaction (PCR) using stool samples is the best method for detecting *E. histolytica* infection [11]; however, it is expensive, time-consuming and requires complicated procedures and is thus not ideal as a screening method. Moreover, the handling of stool samples at voluntary counselling and testing centres in developed countries is inconvenient; most sexually transmitted infection (STI) screening tests at these centres are performed using blood samples. Although the screening utility of serology for asymptomatic *E. histolytica* infected carriers has not been assessed in previous studies, our recent data strongly suggest that serological testing is highly sensitive for detecting symptomatic infectious diseases and asymptomatic *E. histolytica* infection [14].

Here, we prospectively performed serological testing for HIV-negative men who have sex with men (MSM) and confirmed *E. histolytica* infection by PCR for those with positive serology. We also assessed the utility of serological screening for the identification of asymptomatic *E. histolytica* infection.

## Methods

### Ethics statement

This study was approved by the ethics committee of the Center (NCGM-G-002091-00), and all participants provided written informed consent in accordance with the Declaration of Helsinki. All participants gave written informed consent for the study.

### Study population

This cross-sectional study was carried out between January and March 2021 in an HIV-negative MSM cohort at the Sexual Health Clinic of the National Center for Global Health and Medicine (NCGM) [15]. This cohort was a single-centre prospective study. It was established to perform HIV screening and serological testing for syphilis and rectal *Chlamydia trachomatis* and *Neiserria gonorrhoeae* every 3 months for HIV-negative MSM in 2017. Inclusion criteria of the HIV-negative cohort were MSM, aged ≥16 years old, those who have anal sexual intercourse. People diagnosed with HIV at enrolment were excluded from the cohort and were referred to an HIV-positive clinic, the AIDS Clinical Center at NCGM, or other medical institutions.

### Sample size estimation

Sample size estimation to assess the seropositivity of *E. histolytica* among a sexually active MSM population in the study site was performed using Power Analysis & Sample Size 2021 (NCSS Statistical Software, LLC, Utah, USA). The minimum number of necessary samples was estimated as 255 participants. The following numbers were used for the calculation: confidence level 95%, precision, half width 5%, population proportion 21.3%, and population size 24,452 people. The population proportion was estimated using the previously reported seropositivity of *E. histolytica* among HIV-positive MSM [12]. The population size (MSM at study location) was calculated based on the data of 1.2% of Japanese males having sex with men during their life span [16], and that of 2,037,693 males between 21 and 50 years old living in the metropolitan Tokyo area (https://www.toukei.metro.tokyo.lg.jp/juukiy/2021/jy21q10601.htm#kubu).

### Serum anti-E. histolytica testing

The presence of anti-*E. histolytica* antibody was detected using a commercially available ELISA kit (*Entamoeba histolytica* IgG-ELISA; GenWay Biotech, Inc., San Diego, CA. USA). All procedures were performed according to the manufacturer's instructions. In brief, diluted serum samples (100X dilution in IgG sample diluent) as well as 5 control samples, consisting of 1 substrate blank, 1 negative control, 2 cut-off controls, and 1 positive control, were applied to 96-well plates pre-treated with *E. histolytica* antigen and incubated at 37˚C for 1 hour. After washing the plates using washing solution, 100 μL of *E. histolytica* Protein A conjugate was added to all wells except the substrate blank and incubated for 30 minutes in the dark. After a second wash, TMB (3,3',5,5'-Tetramethylbenzidine) substrate solution was added to all wells. After a 15-minute incubation, 100 μL of stop solution was applied to the plates, and absorbance of the specimen was then read at 450/620 nm using a spectrometer. The ELISA titer was calculated by employing correction to obtain the cut-off value [formula used for the

correction: units = (sample absorbance value × 10) / (cut-off absorbance value)]. Positive results were interpreted as 11 units or higher.

### Identification of Entamoeba from stool samples

For seropositive participants, stool samples were obtained and examined by stool ova and parasite examination (O&P), which consisted of direct microscopic examination for trophozoites and formalin-ether sedimentation for cyst forms stained with iodine. A single-round conventional PCR (cPCR) assay for the detection of three *Entamoeba* species (*E. histolytica*, *E. dispar*, and *E. moshkovskii*) was carried out. Stool specimens (approximately 0.2 g) were weighed and subjected to DNA extraction using a QIAamp Fast DNA Stool Mini Kit (Qiagen, Hilden, Germany). DNA extraction was performed according to the manufacturer's instructions. The DNA was eluted in 100 μL of elution buffer (Qiagen) and stored at −80˚C until further analysis. The primer set was designed based on signature sequences in the small-subunit rRNA of each species, of which the utility was confirmed in a previous study [17]. The primer set consisted of the same forward primer (EntaF, 5′-ATGCACGAGAGCGAAAGCAT-3′) in combination with three reverse primers, one for each of the three species (EhR, 5′-GATCTAGAAAC AATGCTTCTCT-3′; EdR, 5′-CACCACTTACTACC-3′; EmR, 5′-CACCACCACTTACT ATCCCTACC-3′). *Entamoeba* species were differentiated based on the sizes of the PCR products (a 166-bp PCR product for *E. histolytica*, a 752-bp PCR product for *E. dispar*, and a 580-bp PCR product for *E. moshkovskii*). Finally, the results were confirmed by DNA sequencing. Sanger sequencing was performed with a BigDye Terminator v3.1 Cycle Sequencing kit (Thermo Fisher Scientific Inc., Tokyo, Japan), and then analysed on an ABI 3730xl DNA Analyzer (Thermo Fisher Scientific Inc., Tokyo, Japan).

### Measurements of other STI testing

Hepatitis B surface antigen, core antibody, and hepatitis C antibody were tested by a chemiluminescent enzyme immunoassay (CLEIA)-based HISCL analyser with HISCL kits (Sysmex Corp. Japan). Serum rapid plasma reagin test (RPR) ["Sankoh" (EIDIA Co, Tokyo)] and *Treponema pallidum* latex hemagglutination assay (TPHA) were performed. The diagnosis of syphilis was based on serum RPR ≥8 and positive TPHA results. A nucleic acid amplification test (Bio Medical Laboratories, Inc., Tokyo, Japan) was used to detect *Chlamydia trachomatis* and *Neiserria gonorrhoeae*.

### Statistical analyses

Comparisons of the qualitative data were carried out with the Chi-square test, and analysis of variance (ANOVA) was used for comparisons of quantitative data. Statistical significance was defined as a two-sided P value < 0.05. All statistical analyses were performed using GraphPad Prism 7.0 (GraphPad Software, Inc., San Diego, CA, USA).

## Results

In total, serological testing for *E. histolytica* was performed for 312 asymptomatic HIV-negative MSM (Fig 1). Of these, 91.3% had only male-to-male sexual contact, while the other 8.7% had bisexual contact (Table 1 and S1 Data). More than half of the participants (158/312) had experienced STIs prior to the present study, although none had a history of treatment for *E. histolytica* infection based on a medical self-declaration form. The overall seropositivity of *E. histolytica* was 6.7% (21/312) (Fig 2A and S2 Data). This was the same positivity as found by RPR testing 5.4% (17/312), in which only four people showed high RPR titres (R.U. > 16.0).

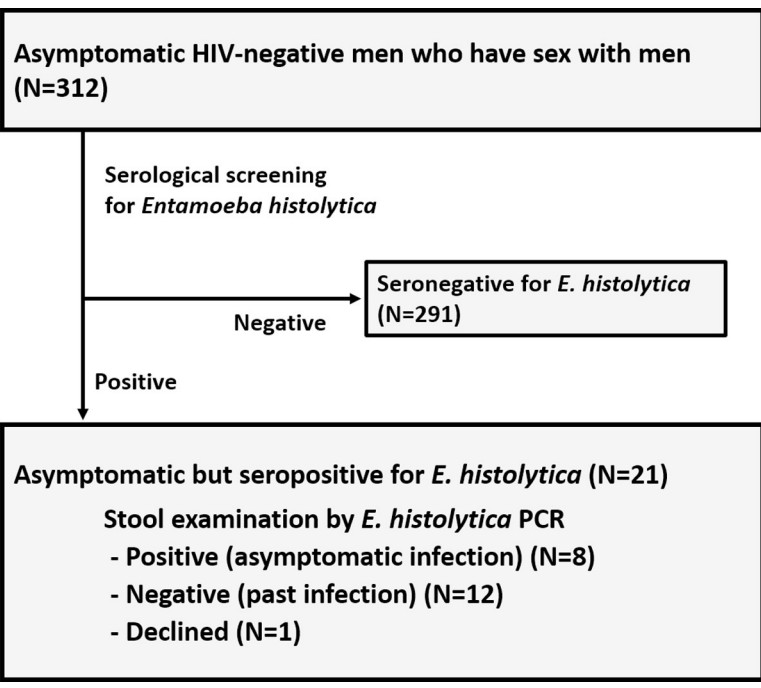

**Fig 1. Study workflow.**

Additionally, *E. histolytica* seropositivity was significantly higher than that of hepatitis B surface antigen and hepatitis C virus antibody. The *E. histolytica* seropositivity was positively correlated with age (Fig 2B). There was no significant correlation between the *E. histolytica*

**Table 1. Characteristics of study participants undergoing a screening test for anti-*E. histolytica* antibody.**

| Median [IQR] or % (N) | All (N = 312) |
|---|---|
| Age | 34 [28–41] |
| Sexual partners | |
| Male only | 91.3% (282/309) |
| Male and female | 8.7% (27/309) |
| Insertive/receptive | |
| Insertive only | 19.5% (60/308) |
| Receptive only | 26.9% (83/308) |
| Both | 51.0% (157/308) |
| No insertive sex | 2.3% (8/308) |
| Number of sexual partners within 6 months | 5 [3–10] |
| Condom use (%) | 60 [20–90] |
| Past treatment of any STIs | 50.6% (158/312) * |
| Past treatment of amebiasis | 0% |

Abbreviations: IQR, inter quartile range; N, number; STIs sexually transmitted infections; AmebaAb, anti-*Entamoeba histolytica* antibody.

*List of past STIs consisted of syphilis (n = 51 cases), *Chlamydia trachomatis* infection (n = 40 cases), condyloma acuminata (n = 33 cases), *Neiserria gonorrhoeae* (n = 26 cases), hepatitis B virus infection (n = 22 cases), pubic lice (n = 22 cases), genital herpes infection (n = 11 cases), hepatitis A virus infection (n = 4 cases), *Mycoplasma genitalium* infection (n = 2 cases), and giardiasis (n = 2 cases) (S3 Data).

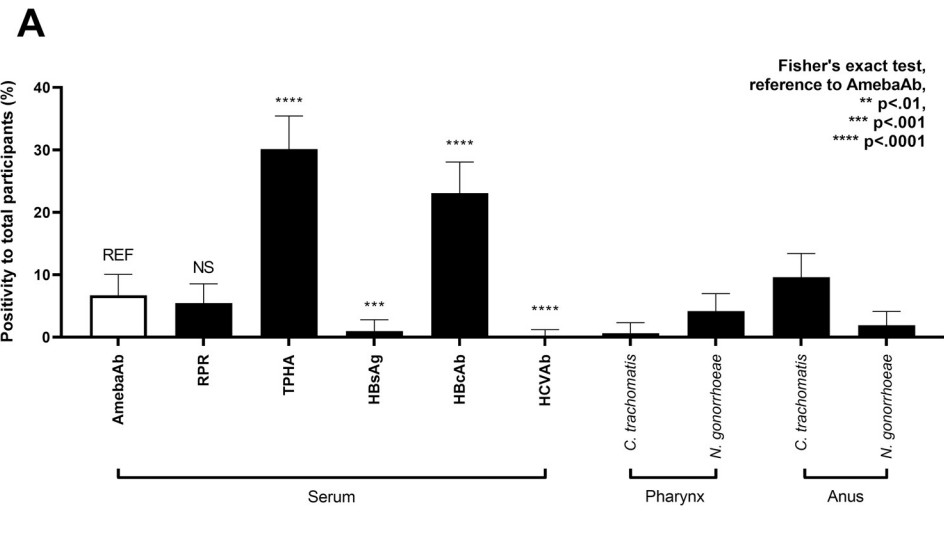

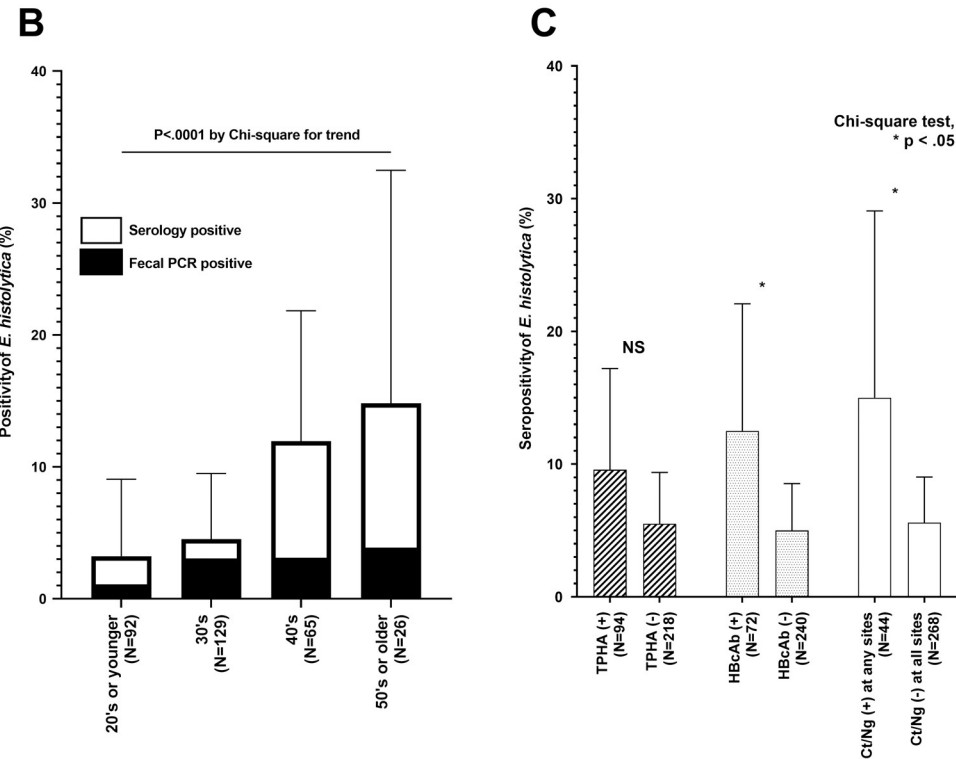

**Fig 2. *Entamoeba histolytica* seropositivity and screening results of other sexually transmitted infections.** (A) Seropositivity of sexually transmitted infections (solid bars) compared with that of *Entamoeba histolytica* (clear bar) by Fisher's exact test. (B) *E. histolytica* seropositivity by age group. The ratio of PCR-positive cases to seropositive cases is indicated by the solid bar. (C) *E. histolytica* seropositivity in those with and without other sexually transmitted infections. Error bars indicate 95% confidence intervals. Abbreviations: AmebaAb, anti-*Entamoeba histolytica* antibody; RPR, rapid plasma regain; TPHA, *Treponema pallidum* hemagglutination; HBsAg, hepatitis B surface antigen; HBcAb, hepatitis B core antibody; HCVAb, hepatitis C virus antibody; PCR, polymerase chain reaction; Ct, *Chlamydia trachomatis*; Ng, *Neisseria gonorrhoeae*; REF, reference data; NS, not significant.

seropositivity and sexual preferences of participants (S2 Data). As expected, *E. histolytica* sero-positivity was relatively high among participants with positive serology for *Treponema palli-dum* hemagglutination or hepatitis B core antibody and among those with *Chlamydia trachomatis* and/or *Neisseria gonorrhoeae* infection (Fig 2C).

Next, to assess current asymptomatic infections among individuals who were seropositive for *E. histolytica*, we performed O&P and PCR of the stool samples to identify the pathogen. One of the 21 seropositive individuals refused stool examination; therefore, we examined a total of 20 stool samples. None of these seropositive individuals had abdominal symptoms at the time of stool sampling. O&P identified cysts in 20% (4/20) of the seropositive participants (cysts in three cases and cysts and trophozoites in one case). PCR identified *E. histolytica* DNA in 40.0% (8/20) of the seropositive participants. Interestingly, one person with cysts had a negative PCR result; this was concluded to be a false-positive by O&P. Thus, we finally identified 8 cases of asymptomatic *E. histolytica* infection in 20 seropositive participants of the 312 HIV-negative MSM cohort.

## Discussion

In the present study, serological testing to identify *E. histolytica* infection was performed for HIV-negative MSM individuals. The overall seropositivity (6.7%) was between that found in HIV-positive individuals (21.3%) [12] and that at a voluntary counselling and testing centre in Tokyo (2.6%) [13], even though no participants had a previous treatment history of *E. histolytica* infection at inclusion. This indicates that *E. histolytica* infection is a common STI among HIV-negative MSM individuals. We also identified current *E. histolytica* infection in 40% of seropositive individuals. This finding is consistent with a previous study that found a 38.9% (7/18) serological testing specificity against colonoscopically identified asymptomatic amoebic colitis [17]. On the basis of the results calculated by dividing 1 by the positive ratio of each test, 37.1 serologic tests followed by 2.5 stool PCR tests were required for the identification of one case of asymptomatic infection in this study population. This is the first study showing that mass-screening by serology can identify new cases of asymptomatic *E. histolytica* infection in a high-risk population.

There are some limitations to this study. First, we performed stool PCR testing only for seropositive participants because of limited funding. We were unable to assess serology false-negatives; however, our previous study using colonoscopy found that the false-negative rate is low (1.9%, 1/53) [18]. The sensitivity and specificity of serological testing to identify asymptomatic infection (effectiveness of the serological screening strategy for asymptomatic *E. histolytica* infection) should be confirmed by a future prospective analysis study, which performs stool PCR and serology for all participants. Second, owing to the small sample size, the epidemiological impact of the applied screening strategy for identifying asymptomatically infected individuals could not be assessed. Our serological screening approach provides a potential strategy for the epidemiological control of re-emerging sexually transmitted *E. histolytica* infection. However, active epidemiological surveys to identify high-risk populations are also essential for the future epidemiological control of sexually transmitted *E. histolytica* infection.

In conclusion, we identified eight patients with *E. histolytica* infection from 312 asymptomatic HIV-negative MSM individuals by serological screening. Active epidemiological surveys, in combination with an effective screening strategy to identify asymptomatically infected individuals, should be considered for the future control of this re-emerging communicable disease.

## Supporting information

**S1 Data. Comparison of characteristics between antibody positive and negative partici-pants.**
(DOCX)

**S2 Data. *E. histolytica* seropositivity and sexual preferences of participants.** There were no significant correlations between the seropositivity and sexual preferences by Fisher's exaxt test or ANOVA test. Error bars indicate 95% confidence intervals. Abbreviations: E. histolytica, Entamoeba histolytica; STI, sexually transmitted infection; Tx, treatment history; NS, not significant.
(TIF)

**S3 Data. Data set of characteristics and test results of study participants.**
(XLSX)

## Acknowledgments

We thank all of the medical staff who assisted with sample collection at the National Center for Global Health and Medicine Sexual Health Clinic. We also thank Katherine Thieltges and J. Ludovic Croxford, PhD, from Edanz (https://jp.edanz.com/ac) for editing a draft of this manuscript.

## Author Contributions

**Conceptualization:** Koji Watanabe.

**Data curation:** Misao Takano, Takahiro Aoki, Daisuke Mizushima, Kenji Yagita.

**Formal analysis:** Yasuaki Yanagawa, Rieko Shimogawara, Kenji Yagita, Koji Watanabe.

**Funding acquisition:** Koji Watanabe.

**Investigation:** Yasuaki Yanagawa, Rieko Shimogawara, Kenji Yagita, Koji Watanabe.

**Methodology:** Rieko Shimogawara, Koji Watanabe.

**Project administration:** Koji Watanabe.

**Resources:** Misao Takano, Takahiro Aoki, Daisuke Mizushima.

**Supervision:** Hiroyuki Gatanaga, Yoshimi Kikuchi, Shinichi Oka, Kenji Yagita, Koji Watanabe.

**Visualization:** Yasuaki Yanagawa, Koji Watanabe.

**Writing – original draft:** Yasuaki Yanagawa.

**Writing – review & editing:** Hiroyuki Gatanaga, Yoshimi Kikuchi, Shinichi Oka, Kenji Yagita, Koji Watanabe.

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
