## [Decision Letter · Decision Letter 0]

8 Dec 2021

Dear Dr. Watanabe,

Thank you very much for submitting your manuscript "Effectiveness of serological testing to detect asymptomatic Entamoeba histolytica infection: A cross-sectional study of an HIV-negative men who have sex with men cohort in Japan" for consideration at PLOS Neglected Tropical Diseases. As with all papers reviewed by the journal, your manuscript was reviewed by members of the editorial board and by several independent reviewers. In light of the reviews (below this email), we would like to invite the resubmission of a significantly-revised version that takes into account the reviewers' comments. 

We cannot make any decision about publication until we have seen the revised manuscript and your response to the reviewers' comments. Your revised manuscript is also likely to be sent to reviewers for further evaluation.

Sincerely,

Aysegul Taylan Ozkan, Ph.D., M.D.

Deputy Editor

Aysegul Taylan Ozkan

Deputy Editor

Reviewer's Responses to Questions

**Key Review Criteria Required for Acceptance?**

**Methods**

-Are the objectives of the study clearly articulated with a clear testable hypothesis stated?

-Is the study design appropriate to address the stated objectives?

-Is the population clearly described and appropriate for the hypothesis being tested?

-Is the sample size sufficient to ensure adequate power to address the hypothesis being tested?

-Were correct statistical analysis used to support conclusions?

-Are there concerns about ethical or regulatory requirements being met?

Reviewer #1: (No Response)

Reviewer #2: This is an descriptive study for MSM people where the authours have found a seropositive for E. histolytica . 40% of the seropositive patients also positive for E. histolytica in their stool samples

L-117,118 “In addition to stool ova and parasite examination (O&P)”- This line the authours has to clarify

L-123 “eluted in 100 mL of elution buffer”- pleasae acheck this line 

L-131,132 “Finally, the results were confirmed by DNA sequencing”- should include briefly the procedures or a reference of the method..

**Results**

-Does the analysis presented match the analysis plan?

-Are the results clearly and completely presented?

-Are the figures (Tables, Images) of sufficient quality for clarity?

Reviewer #1: (No Response)

Reviewer #2: L-143 “none had a history of E. histolytica infection”- How did the author confirm this? There is no precise statement.

L-147 “hepatitis B surface antigen and hepatitis C virus antibody”- Which methods were utilized to find those? Should include in methods section.

L-175,176 “One of the 21 seropositive individuals refused stool examination”- therefore, author examined a total of 20 stool samples. In this case, PCR detected 40% E. histolytica, yet the author computed all 21 samples in table 1 –should it be revised

**Conclusions**

-Are the conclusions supported by the data presented?

-Are the limitations of analysis clearly described?

-Do the authors discuss how these data can be helpful to advance our understanding of the topic under study?

-Is public health relevance addressed?

Reviewer #1: (No Response)

Reviewer #2: The manuscript does not provide any justification how E. histolytica infections correlate with STI’s. 

Since this study focused on active epidemiological surveys, the author should specify the number of people that tested positive for rapid plasma reagin (RPR) out of the 21 Eh seropositive samples.

**Editorial and Data Presentation Modifications?**

Reviewer #1: In Table 1, the proportions of individuals with or without antibody should be shown in each category. For example, in male only individuals as sexual partner, the proportion of antibody (+) should be calculated as 20/282 (7.1%) and that of antibody (-) should be 262/282 (92.9%). In individuals with male and female as sexual partner, he proportion of antibody (+) should be calculated as 1/27 (3.7%) and that of antibody (-) should be 26/27 (96.3%). As such, I request to recalculate the proportion of antibody (+) and (-) with making the total number of each category as a denominator.

Reviewer #2: (No Response)

**Summary and General Comments**

Reviewer #1: This study showed the effectiveness of serological test in screening asymptomatically infected individuals with Entamoeba histolytica in HIV-negative men who have sex with men in Japan as a cross-sectional study. The study is meaningful, but there are several points to be clarified or improved.

Major points

It is recommended to clearly describe the background.

1) Would you describe why HIV-negative men who have sex with men (MEM) can be a target for the screening of asymptomatic Entamoeba histolytica infection, in addition to the reason that “sexually active individuals”?

2) How was the cohort of HIV-negative MEM set? Can you show the outline of this cohort?

3) Do 312 participants represent all of HIV-negative MEM from the cohort? If 312 participants are a part of the cohort, how did the authors select 312 HIV-negative MEM from the cohort? 

4) In Table 1, the proportions of individuals with or without antibody should be shown in each category. For example, in male only individuals as sexual partner, the proportion of antibody (+) should be calculated as 20/282 (7.1%) and that of antibody (-) should be 262/282 (92.9%). In individuals with male and female as sexual partner, he proportion of antibody (+) should be calculated as 1/27 (3.7%) and that of antibody (-) should be 26/27 (96.3%). As such, I request to recalculate the proportion of antibody (+) and (-) with making the total number of each category as a denominator.

Minor points

1) Line 102: Would you describe the name of E. histolytica antigen(s) and the concentration of the pre-coated antigen(s)? 

2) Line 117: Would you describe “stool ova and parasite examination” more accurately or scientifically in detail?

3) Line 132: Would you describe the methods and results of the DNA sequencing, if the PCR products were confirmed by sequencing?

4) Line 143: Would you describe what kinds of STIs the 159 participants experienced?

5) Line 150-: Would you describe the abbreviations you described in the Fig 2B, like Treponema pallidum hemagglutination (TPHA) etc? The name of species should write down in Italic. 

6) Line 152: Whould you clarify how the participants were “affected”?

7) Line 181-184: It is not easy to understand the meaning of the sentence.

Would you edit “to identify asymptomatic E. histolytica infection”  “to identify at least an asymptomatic E. histolytica infection”

Reviewer #2: Overall this is a nice story on asymptomatic E. histolytica infection in mem sex with men.

PLOS authors have the option to publish the peer review history of their article (what does this mean?). If published, this will include your full peer review and any attached files.

Reviewer #1: No

Reviewer #2: Yes: Rashidul Haque
---

## [Decision Letter · Decision Letter 1]

15 Feb 2022

Dear Dr. Watanabe,

Thank you very much for submitting your manuscript "Effectiveness of serological testing to detect asymptomatic Entamoeba histolytica infection: A cross-sectional study of an HIV-negative men who have sex with men cohort in Japan" for consideration at PLOS Neglected Tropical Diseases. As with all papers reviewed by the journal, your manuscript was reviewed by members of the editorial board and by several independent reviewers. In light of the reviews (below this email), we would like to invite the resubmission of a significantly-revised version that takes into account the reviewers' comments. 

We cannot make any decision about publication until we have seen the revised manuscript and your response to the reviewers' comments. Your revised manuscript is also likely to be sent to reviewers for further evaluation.

Sincerely,

Aysegul Taylan Ozkan, Ph.D., M.D.

Deputy Editor

Aysegul Taylan Ozkan

Deputy Editor

Reviewer's Responses to Questions

**Key Review Criteria Required for Acceptance?**

**Methods**

-Are the objectives of the study clearly articulated with a clear testable hypothesis stated?

-Is the study design appropriate to address the stated objectives?

-Is the population clearly described and appropriate for the hypothesis being tested?

-Is the sample size sufficient to ensure adequate power to address the hypothesis being tested?

-Were correct statistical analysis used to support conclusions?

-Are there concerns about ethical or regulatory requirements being met?

Reviewer #1: (No Response)

Reviewer #3: 1. While the data may be interesting from the perspectives of E. histolytica infection remaining prevalent among MSM in developed countries, particularly in East Asia, there are major concerns regarding the study design.

2. The major weakness of this study was that only the participants who were seropositive for E. histolytica underwent PCR assay to identify intestinal infection with E. histolytica. Given the high sensitivity and specificity of PCR assay for detection of E. histolytica, PCR assay should be considered as gold standard for the diagnosis of intestinal infection with E. histolytica. All recruited subjects should undergo PCR assay of stool samples, followed by serologic assay to better understand the performance of serologic assay used in this study. With the understanding of the performance of serologic assay could the authors be able to examine the “effectiveness” of serologic assay in identifying high-risk individuals with E. histolytica infection. It is understandable that serologic screening would be cheaper and more simple and convenient to perform than PCR assay of stool samples from clinicians’ standpoint; however, to better examine the “effectiveness” or “cost-effectiveness” of serologic testing that is to be widely used for screening in the clinical settings, both PCR assay of stool samples and serologic testing of all recruited participants, not selected participants, should be performed side by side. While the authors discussed it as a limitation, the last several sentences (Lines 218-221) in Results and Conclusions could be incorrect without testing all participants with the use of PCR assay.

3. The authors are encouraged to provide the sample size estimation. It is not clear that how many MSM had been recruited for STI studies at the voluntary counseling and testing site and how many agreed to participate in serologic and PCR testing in this cross-sectional survey.

4. While testing for gonorrhea, syphilis and chlamydia was performed every 3 months, it sounds that serologic testing for E. histolytica was only performed at baseline. Did the authors follow the participants using the same serologic assay to estimate the seroconversion rate?

**Results**

-Does the analysis presented match the analysis plan?

-Are the results clearly and completely presented?

-Are the figures (Tables, Images) of sufficient quality for clarity?

Reviewer #1: (No Response)

Reviewer #3: 1. In Table 1, 3 out of 311 participants had had treatment of amebiasis. In the footnote, 4 individuals had had amebiasis as past sexually transmitted infections (STIs). The authors are encouraged to provide more information on these participants who either had had treatment or amebiasis as STIs because none were reported to have had previous E. histolytica infection in Results section. Moreover, the statement (Lines 227-228) in the first paragraph of Discussion is contradictory to the data presented in Table 1. 

2. Were the ELISA units of the participants testing positive for E. histolytica by PCR assay higher than those of participants testing seropositive but negative by PCR assay?

3. Table 1 could be improved by providing data of “all participants”, “E. histolytica-seropositive participants”, and “E. histolytica-seronegative participants”, with p-values for the comparisons between the latter two groups.

4. In Table 1, should “number of sex within 6 months” be “number of sexual partners within 6 months”?

**Conclusions**

-Are the conclusions supported by the data presented?

-Are the limitations of analysis clearly described?

-Do the authors discuss how these data can be helpful to advance our understanding of the topic under study?

-Is public health relevance addressed?

Reviewer #1: (No Response)

Reviewer #3: While the authors discussed it as a limitation, the last several sentences (Lines 218-221) in Results and Conclusions could be incorrect without testing all participants with the use of PCR assay.

**Editorial and Data Presentation Modifications?**

Reviewer #1: (No Response)

Reviewer #3: (No Response)

**Summary and General Comments**

Reviewer #1: (No Response)

Reviewer #3: The authors performed a cross-sectional serologic survey of E. histolytica infection among 312 HIV-negative men who have sex with men (MSM) and had had no known previous history of E. histolytica infection in Tokyo. Polymerase-chain-reaction assay was performed only in those 20 individuals who tested seropositive for E. histolytica, in which 8 (40%) tested positive for E. histolytica, suggesting current infection. While the data may be interesting from the perspectives of E. histolytica infection remaining prevalent among MSM in developed countries, particularly in East Asia, there are major concerns regarding the study design.

PLOS authors have the option to publish the peer review history of their article (what does this mean?). If published, this will include your full peer review and any attached files.

Reviewer #1: No

Reviewer #3: No
---

## [Decision Letter · Decision Letter 2]

27 Mar 2022

Dear Dr. Watanabe,

Thank you very much for submitting your manuscript "Identification of asymptomatic Entamoeba histolytica infection by a serological screening test: A cross-sectional study of an HIV-negative men who have sex with men cohort in Japan" for consideration at PLOS Neglected Tropical Diseases. As with all papers reviewed by the journal, your manuscript was reviewed by members of the editorial board and by several independent reviewers. The reviewers appreciated the attention to an important topic. Based on the reviews, we are likely to accept this manuscript for publication, providing that you modify the manuscript according to the review recommendations. 

Sincerely,

Aysegul Taylan Ozkan, Ph.D., M.D.

Deputy Editor

Aysegul Taylan Ozkan

Deputy Editor

Reviewer's Responses to Questions

**Key Review Criteria Required for Acceptance?**

**Methods**

-Are the objectives of the study clearly articulated with a clear testable hypothesis stated?

-Is the study design appropriate to address the stated objectives?

-Is the population clearly described and appropriate for the hypothesis being tested?

-Is the sample size sufficient to ensure adequate power to address the hypothesis being tested?

-Were correct statistical analysis used to support conclusions?

-Are there concerns about ethical or regulatory requirements being met?

Reviewer #1: Fine

Reviewer #2: OK

Reviewer #3: The revision made in response to previous comments and queries is acceptable.

**Results**

-Does the analysis presented match the analysis plan?

-Are the results clearly and completely presented?

-Are the figures (Tables, Images) of sufficient quality for clarity?

Reviewer #1: Fine

Reviewer #2: OK, Authors can take out the last sentence of the Results section and use it in the discussion section

Reviewer #3: The revision made in response to previous comments and queries is acceptable.

**Conclusions**

-Are the conclusions supported by the data presented?

-Are the limitations of analysis clearly described?

-Do the authors discuss how these data can be helpful to advance our understanding of the topic under study?

-Is public health relevance addressed?

Reviewer #1: Fine

Reviewer #2: OK

Reviewer #3: The revision made in response to previous comments and queries is acceptable.

**Editorial and Data Presentation Modifications?**

Reviewer #1: No editorial suggestion

Reviewer #2: OK

Reviewer #3: (No Response)

**Summary and General Comments**

Reviewer #1: No additional comment

Reviewer #2: The last sentence of the abstract section is not required authors may take it out. This sentence is also included in the result section but, authors can take it to the Discussion section

Reviewer #3: The authors have responded to the queries and comments raised in the second round of review. While there are weaknesses and limitations of the study design, the authors have revised the manuscript as much as they can in providing the data on the seroprevalence of E. histolytica infection among HIV-negative men who have sex with men. The results of the study have important clinical and public health implications when it comes to prevent transmission of E. histolytica infection among the at-risk population in a developed conutry.

PLOS authors have the option to publish the peer review history of their article (what does this mean?). If published, this will include your full peer review and any attached files.

Reviewer #1: No

Reviewer #2: Yes: Rashidul Haque, icddr,b, Dhaka, Bangladesh

Reviewer #3: No

Figure Files:

Data Requirements:

Reproducibility:

References

---

## [Editor Report · Decision Letter 3]

3 Apr 2022

Dear Dr. Watanabe,

We are pleased to inform you that your manuscript 'Identification of asymptomatic Entamoeba histolytica infection by a serological screening test: A cross-sectional study of an HIV-negative men who have sex with men cohort in Japan' has been provisionally accepted for publication in PLOS Neglected Tropical Diseases.

Best regards,

Aysegul Taylan Ozkan, Ph.D., M.D.

Deputy Editor

Aysegul Taylan Ozkan

Deputy Editor

---

## [Editor Report · Acceptance letter]

21 Apr 2022

Dear Dr. Watanabe,

We are delighted to inform you that your manuscript, "Identification of asymptomatic Entamoeba histolytica infection by a serological screening test: A cross-sectional study of an HIV-negative men who have sex with men cohort in Japan," has been formally accepted for publication in PLOS Neglected Tropical Diseases.

Best regards,

Shaden Kamhawi

co-Editor-in-Chief

Paul Brindley

co-Editor-in-Chief
